# Characterization of the Morphological and Chemical Profile of Different Families of Microplastics in Samples of Breathable Air

**DOI:** 10.3390/molecules28031042

**Published:** 2023-01-20

**Authors:** Joaquín Hernández-Fernández, Esneyder Puello-Polo, John R. Castro-Suarez

**Affiliations:** 1Chemistry Program, Department of Natural and Exact Sciences, San Pablo Campus, University of Cartgena, Cartagena 130015, Colombia; 2Chemical Engineering Program, School of Engineering, Universidad Tecnológica de Bolivar, Parque Industrial y Tecnológico Carlos Vélez Pombo Km 1 Vía Turbaco, Cartagena 130001, Colombia; 3Department of Natural and Exact Science, Universidad de la Costa, Barranquilla 080002, Colombia; 4Grupo de Investigación en Oxi, Hidrotratamiento Catalítico y Nuevos Materiales, Programa de Química-Ciencias Básicas, Universidad del Atlántico, Puerto Colombia 081001, Colombia; 5Área Básicas Exactas, Universidad del Sinú, Seccional Cartagena, Cartagena 130001, Colombia

**Keywords:** microplastics, respirable air, morphological profile, pollution, DSC, Pyr-GC/MS

## Abstract

Microplastic (MP) contamination has become a problem of great interest to the community at large. The detection of these particles in different ecosystems and foods has been the subject of study. However, the focus of these investigations has been on the identification and quantification of PM by DSC and Pyr-GC/MS and not on how they are transported to reach the air we breathe. In this study, the values of morphological parameters for plastic particles in a range between 1 and 2000 µm, present in the breathable air of 20 neighborhoods in the city of Cartagena, Colombia, were obtained to determine the characteristics that make these particles airborne. The values of parameters were obtained, such as roundness, sphericity, curvature, and the convexity of the particle, as well as its compactness and size, which influence its transport through the air and its ability to be ingested and inhaled. The data obtained in this study allows for simulations and the analysis of the behavior of microplastics once in the environment to predict future settlements. The DSC showed us the melting temperatures of PP, PE, PET, and PS, the Pyr-GC/MS showed the fragmentation patterns, and the presence of these MPs in the samples was confirmed.

## 1. Introduction

In recent years, microplastics have been identified and quantified in different sources that represent a potential threat to the health of humanity due to the increased levels of MP in the air, water, animals, and food that we consume on a daily basis [1,2,3]. The detection, quantification, and identification of microplastics in different places have made it possible to establish the risks to which we are exposed due to the decomposition of plastic materials used in different aspects of daily life [4,5,6]. Because of their small size, microplastics (MP) are easily transported through the environment and ingested through respiration or consumed food [7,8]. All these aspects have increased the interest in the study of MP, but the main focus has been on detection and identification and not on studying how the microstructure of these materials can affect their transport and intake.

The study of the morphology of microplastics, their distribution in different parts of a city, and the identification of the type of plastic allow us to understand how it is transported, which in turn, helps to establish methodologies and techniques that allow the mitigation of these MP in the air, which we breathe in various parts of town [9,10]. The distribution and settling of microplastics are determined by different physical and environmental factors that allow the transport of these particles through the environment [11,12]. To understand how MPs interact in the environment either to be transported, deformed by friction effects with the air and/or objects, and to be ingested and circulated in the human body, it is necessary to know several morphological parameters that allow predictive studies of their behavior [13,14]. However, due to the focus that studies on these particles generally receive, information on their physical characteristics has been limited, and it is difficult to know the necessary parameters.

MPs can be identified by means of various analytical techniques that allow one to know the material that constitutes them, including the additives present in them, which can be hazardous to health [15,16]. Among the techniques for the identification of microplastics are the differential scanning scanner (DSC) [17] and Fourier transform infrared spectroscopy (FTIR), which is a non-destructive technique and allows the composition of the studied material to be known [18]; alongside this, pyrolysis together with gas chromatography-mass spectrometry (Pyr-GC/MS) by the thermal degradation of the material and analysis of the resulting compounds allows the identification of the material [19]. For the morphological analysis and obtaining of physical parameters of microplastics, there are not many records. However, the techniques used for the characterization of particles that implement images and software and that allow the physical parameters of interest to be obtained can be implemented [20].

Microplastics have been identified in different countries and by different techniques. In Cartagena, Colombia, there is a whole industrial zone where a considerable number of plastic-producing plants (petrochemical plants) are located that can contribute to the presence of microplastics and pollutants in the environment and which can affect the health of the surrounding inhabitants [21,22,23,24,25,26,27,28,29,30,31,32,33,34,35,36,37,38,39]. To identify how they can be dispersed and which would be the populations or sites most affected by MPs, it is necessary to identify and obtain the morphological parameters of the particles.

In this research, breathable air samples are taken in 20 neighborhoods of the city of Cartagena, Colombia, to analyze their characteristics, classifying them by size, color, and type of plastic, in addition to obtaining morphological parameters that influence the movement and transport of MPs in the environment. With these, we can develop predictive models of possible settlements and identify affected areas and the incidence of plastic types in the formation of MPs.

## 2. Results and Discussion

### 2.1. Morphological Characteristics

The microplastic particles were classified by color in the different neighborhoods. The colors identified were white, black, blue, yellow, red, green, and, in small proportions, other colors such as brown and transparent. The colors that were used in greater proportion were black and white in all the neighborhoods, as seen in Figure 1. The colors of the microparticles can also determine their origin. However, this must be accompanied by the type of plastic to identify whether it comes from clothing, toys, tires, or some other polymer that has degraded. The limitation on color identification is associated with the degradation processes that the material has undergone. In some cases, the color may be a little different from the original due to the oxidation that the material has had over time, making the color more undefined than it should be. The identification of transparent microplastics is more complicated. Those that were identified had a particle size greater than 800 µm, so they could be seen. In the same way, it is not ruled out that some smaller particles were not identified. The transparent color that can be affected by the color of the filter used, the size of the, and the shape, makes it difficult to identify.

The classification by size was carried out in the different samples from 1 to 2000 µm in different intervals, as shown in Figure 2. A greater proportion were microplastics, which, according to the literature, are those particles with a size between 1 and 5 µm. When analyzing the distribution of sizes and comparing it with the location of the neighborhoods, a trend was identified towards the neighborhoods located in the north, where particles with sizes between 1 and 5 µm were found in a greater concentration when approaching this area. The foregoing may be associated with different environments, such as the direction and speed of the wind, which in this city is predominant to the north, with varying speeds between 1 and 7 knots (65 to 80%) and between 7 to 22 knots (4 to 28%) depending on the month. Factors such as population distribution, rainfall, and buildings also influence the transport of particles.

The shape of the microplastic influences the ease of transport, along with the size. The larger the particle, the more difficult it will be to transport by the wind, and depending on its shape, it will define how it can be transported. This is due to the friction it would have with the wind. Table 1 and Figure 3 and Figure 4 show the mean values obtained for the different morphological parameters studied. The particles that presented greater sphericity, compactness, circularity, solidity, and extent were those present in the size ranges 50–100, 100–200, 200–300, 600–700, 700–800, and 1500–2000. This shows that the ranges smaller than 50 microns were not spherical. The transparency of the particles in the entire study size range increased with the decreasing size, showing the range of 1500–2000 at a transparency of 0.051 and the range of 1–5 at a transparency value of 0.68.

The values obtained depend on several factors, such as material, sample size, and the formation and/or origin of the microplastic, among others. Due to these factors, there is no direct relationship between the particle size and the value obtained. It is also important to emphasize that to the south of the city, there are petrochemical plants that produce a variety of plastic materials, which helps to have a higher concentration of microplastics in that area. However, as the neighborhoods move away from the industrial zone, that is, towards the north, the concentration of microplastics decreases, so it can be considered that the proximity to petrochemical plants can potentiate not only the intake of microplastics but also the intake of different components that can affect health.

Various morphological parameters related to the shape of the particles were generally used in the study of particulate material on an industrial scale (sand). However, these parameters also apply to any particle; they can be very useful for the morphological study of microplastics. The morphological parameters of great interest are length, circularity, roundness, sphericity, area, elongation, and cavities, among others. These parameters are determined by factors such as whether the study is carried out in 2D or 3D and whether it is microscale, macroscale, or medium-scale [40,41]. The values obtained in these parameters, together with factors common to climatic effects such as wind speed, particulate material humidity, and the transition regime through which they move, affect the transport of microparticles. For a more complete study of the effects of the morphology of microplastics on their transport, it is necessary to know the aerodynamic diameter of these particles, which is defined as the diameter of a sphere with a standard density that settles at the same terminal velocity as the particle of interest, which can be obtained by different instruments.

However, there are some equations that allow this according to the flow regime in which the particle moves, which can be calculated and determined by the Knudsen number (Kn) as shown in Equation (1), which depends on the mean free path of the gas molecules (λ), the radius (r), or diameter (d). If Kn = 1, the regime is continuous, Kn ≫ 1 indicates that it is a free molecular regime, and if 0.1 ≤ Kn ≤ 10, it is known as the transition regime [42].
Kn = λ/r = 2λ/d,(1)

Equations (2) and (3) show the calculation of the aerodynamic diameter in free flow and parameters such as sphericity [43]. The morphological parameters obtained, together with the density of the identified material, allow us to understand the behavior of microplastics in the environment. However, with a wide spectrum of particle sizes and different polymeric materials with different morphologies, it is difficult to study each one of them. For this, it is necessary to carry out the study with a specific particle size and evaluate a single morphological parameter of the particle at the same time. Precipitation is not considered in these equations since this affects the free movement of particles.
(2)$$d_{a}=(\rho_{p}/\rho_{0})\times \left(d_{e}/X_{v}\right)$$
(3)$$S=\left(\sqrt[{3}]{36\pi V^{2}}\right)/SA$$

*d_a_* = Aerodynamic diameter*ρ_p_* = Density of the particle material*ρ*_0_ = Standard density (1 g cm^−3^)*d_e_* = Volume equivalent to diameter*X_v_* = Dynamic shape factor*S* = Sphericity*V* = Volume of the particles*SA* = Surface area

Microplastics may enter the respiratory or digestive systems by ingestion or inhalation, causing them to accumulate in the various organs of these systems. MPs can be retained in the gastrointestinal system after entering the human body [44]. The diameters and morphological properties allow it to be more or less aerodynamic and determine the depth that the MP remains lodged in the organs, allowing the speed and ease of an MP particle that is kept in any organ to be calculated [2,45]. Smaller particles of MP are more likely to enter the respiratory tract. Plastic fibers were discovered in lung tissue, proving that the fibers may penetrate deep into the lung. Studies have shown that polypropylene and polyethylene fibers can remain in the synthetic lung fluid for 180 days without any variation [46,47,48]. In addition, it has been shown that smaller plastic particles can increase the probability of penetrating the membranes and barriers of the organism. The selectivity of Py-GC-MS enables the analysis of the mixture of MPs, avoiding the need to isolate single particles.

### 2.2. Identification of MPs by DSC

For the identification of the material of the microplastics, they were first classified according to their size and the study neighborhood, as shown in Table 2, where zero represented the fact that the particle size was not identified in that area.

The physicochemical properties of PE, PP, PET, and PS indicate that their melting initiation temperature was 110, 161, 248, and 96 °C, respectively. This information makes it possible to show that in Figure 5 (The thermogram with the blue line refers to a sample of microplastics with a higher concentration of PET, PP, PE, and PS, and the thermogram with the black line was when a sample of microplastics has a lower concentration of PET, PP, PE, and PS. The red line is the slope to determine the area under the curve of each peak), the first, second, third, and fourth peaks correspond to PS, PE, PP, and PET.

The identified plastics are polystyrene (PS), which is used in the production of disposable plates and sound insulation materials, among other things [49]; low-density polyethylene (PE), which is used in personal hygiene products, films for food packaging, some textiles, and so on [50]; polypropylene (PP), can be derived from synthetic fibers and is used in textiles, containers, and so on [51]; and low-density polyethylene (LDPE), is used as a packaging material [52]. These materials have distinct melting temperatures that allow them to be identified and are frequently found in commonly used things, either because they were intended to be that way (such as microplastic spheres for exfoliants) or because of damage, which causes them to degrade and produce microplastics.

### 2.3. Confirmation of Chemical Composition of MPs by Py-GC-MS

#### Calibration

Calibration curves are obtained by pyrolyzing the individual polymers. We worked with sample weights within the calibration ranges of between (0.1 and 1070 µg). The results are summarized in Table 3. The pyrolysis products of PP, PE, PS, and PET (pyrolytic markers) are separated by GC, and the ions generated in the mass spectrometer are separated and quantified. The purpose of greater stability is more characteristic but not common among all the polymers of interest, which are selected as pyrolytic markers to identify one polymer from another. The obtained and associated ions are used for quantification according to the calibration protocols.

Table 3 shows the markers of cad polymers. For PS, the 2,4-diphenyl-1-butene was selected; for the PP the 2,4-dimethyl-1-heptene; for the PE, the n-tetradec-1-ene; and for the PET, the markers were the vinyl benzoate and the divinyl terephthalate. Theoretical Kovats retention indices (IR) were selected from the literature. The RI values for this investigation were calculated using the retention times of compound x, the n alkane with n carbon atoms eluting immediately before compound x, and the n-alkane with n + 1 carbon atoms that elute immediately after compound x. This value is compared with the theoretical RI. These values are shown in Table 3. The calibration cuvette is built by pyrolyzing different amounts of PP, PE, PET, and PS. These tests showed R2 to be greater than 0.992 for 2,4-diphenyl-1-butene, 2,4-dimethyl-1-heptene, and n-tetradec-1-ene. The lowest R2 was for the PTE markers vinyl benzoate and divinyl terephthalate.

Table 4 lists the chemical compounds that were identified when the pyrolysis of the samples containing the mixtures of PET, PP, PE, and PS was carried out. The chromatographic profile shows multiple peaks from different areas, and the most significant and best-resolution ones are listed in Table 4. Each compound was identified with its RI and a group of qualifying ions. The chemical structure of each identified compound is related to the chemical structure of the MPs present, and this allowed them to be associated with a specific polymer. Each compound that was identified with chemical structures or chemical groups in common was grouped into specific groups. In this way, five groups were organized as 1, 2, 3, and 4. Table 4 shows that the chemical compounds of groups 1, 2, 3, and 4 have structures typical of PET, PP, PE, and PS. For example, compounds such as benzene, vinyl benzoate, benzoic acid, diphenyl, divinyl terephthalate, ethanediol dibenzoate, and 2-(benzoyloxy) ethylvinyl terephthalate, identified in the pyrolysis of the samples, are directly related to the structure of the PET macromolecule. For the case of molecules as 2,4,6-trimethyl-1-nonene, 2,4,6,8-tetramethyl-1-undecene, 2,4,6,8,10-pentamethyl-1-tridecene, and

2,4-Dimethyl-1-heptene, which was also identified in the pyrolysis, is uniquely related to the structure of the PP polymer. In Table 4, it is also observed that styrene, α-methylstyrene 1,3-Diphenylpropane, styrene dimer, and styrene trimer were identified during the pyrolysis of the microplastics and this group of aromatic molecules is uniquely related to the PS polymer structure. The n-pentadec-1-ene molecule was identified in the pyrolysis of the obtained microplastics. This molecule is related to the structure of PE. In addition to the above, the n-pentadec-1-ene molecule is a marker of identification for PE, which allows greater reliability. Figure 6 shows the chemical structures of these compounds and the MP that originates from them. Figure 6 shows the condensed structure of the PET, PP, PE, and PS polymers and also of the most representative molecules that were identified after the pyrolysis of microplastics. With the information in Figure 6, we can see that the molecular structure of the chemical species has common functional groups with each microplastic, and in this way, we can clearly see that the origin of each molecule can be identified by mass spectrometry originates from the pyrolysis. of each specific polymer. This Py-GC-MS analysis corroborates the presence of the MPs identified by DCS.

## 3. Materials and Methods

### 3.1. Materials

To avoid contamination of the samples, glassware was used during their collection. All reagents and solutions used during this investigation were treated with distilled water filtered using a cellulose nitrate filter, and only cotton lab coats were used for the researchers’ personal protective equipment. Narrative iron sulfate, 30% peroxide, 99.9999% acetone, and concentrated sulfuric acid (Merck, Darmstadt, Germany) were used.

### 3.2. Place of Sampling; Collection and Treatment of Samples

The research was carried out in 20 neighborhoods of the city of Cartagena de Indias in Colombia, where samples were collected between January 2018 and December 2021. Figure 7 shows the neighborhoods where samples were taken. The temperature during the sampling periods oscillates between 27 °C and 30 °C. The reactive humidity varied between 80 and 83%.

#### Breathing Air Samples

Three people participated in the breathing air sampling. For the collection of these samples, a 50-mL beaker containing 20 mL of deionized water was placed. The passage was at the level of the nose between the nostrils and the chin; the upper surface of the vessel was 5 mm from the nostrils and was completely open (See Figure 8). This procedure was carried out for 5 days in the 20 neighborhoods of interest. During the sampling period, these people did not move to places other than these neighborhoods, except for moving inside their homes and into the surrounding areas of the same neighborhood. The samples collected in each neighborhood were marked and labeled correctly to avoid cross-contamination, resulting in a total of 300 samples. For the extraction of the microplastics, a cellulose nitrate filter with a pore size of 0.45 μm was used, which 50 mL of a Nile red solution (the concentration of the red null prepared was 100 ppm, and acetone was used as the diluting solvent) was added which dyed the microplastics. They were in contact for a period of 30 min for later collection. The filter containing the stained MP samples was dried at 50 °C in an oven and then stored in Petri dishes.

### 3.3. Microplastic Quantification and Morphological Profile

#### 3.3.1. Morphological Profile

The morphological profile was evaluated using an Axiocam MRm digital camera, Dino-Light Edge, Serial: AM4115TL-GFBW, Microtrac S3500 (York, PA 17406, US) and ImageJ software V.1.113. Microplastics were classified according to their geometry, size, and color in the different zones. ImageJ software has been widely used for image analysis in different areas [53,54], as it allows the acquisition of accurate data and the calculation of parameters and country variables [55,56].

#### 3.3.2. Identification of Microplastics

##### Differential Scanning Calorimetry (DSC)

DSC measurements are carried out using a DSC standard cell RC. The melting temperatures of the polymers were utilized to identify them, and the resulting masses were determined based on the fraction of each sample’s aliquot [24].

##### Py-GC-MS

Multiple shot pyrolyzer (EGA/PY-3030D Frontier Lab, Koriyama, Japan) coupled to a gas chromatograph with mass spectrometry (7890B and 5977B Agilent Technology, Santa Clara, CA, USA) was used. The pyrolysis was carried out at 500 °C, and the generated gases passed to a fused silica capillary column of the GC column (HP-5MS, stationary phase 5%-diphenyl, 95%-dimethylpolysiloxane 30 m × 0.25 mm id × 0.25 µm film thickness). Helium was used as a carrier gas to transport the samples to the interior of the chromatographic column at a rate of 1.0 mL min^−1^. The MS detector operated on electron ionization at 70 eV with scans in a range from 35 to 600 m/z. The temperature of the ion source and the interface were 230 °C and 240 °C, respectively. The initial temperature of the oven was 50 °C, with a ramp from 4.2 °C min-1 to 160 °C and a second ramp of 20 °C min^−1^ at 325 °C, where it was maintained for 5 min. The injector temperature was 280 °C. 

The identification of compounds of interest related to fragmentation patterns of PP, PE, and PS was based on the pairing of mass spectra of the NIST 14 database, the scientific literature, analysis, interpretation of the mass spectrum, the pyrolysis of reference compounds, and, in addition, the comparison with Kovats retention indices (IR). For system calibration, specific reference selections of PP, PE, PET, and PS were selected and individually pyrolyzed. The samples were weighed on a Mettler Toledo microbalance (STDA851 TGA, Columbus, OH, USA) with 0.001 mg sensitivity. From the individual pyrolysis, chemical molecules were selected to act as pyrolytic markers for each.

## 4. Conclusions

In this work, microplastics were identified in respirable air in different neighborhoods of the city of Cartagena, Colombia, in order to analyze the morphological profile of these particles and to identify the polymeric material that constitutes them. White, black, blue, yellow, red, and green microplastics and, in small proportions, other colors such as brown and transparent were identified. The morphological analysis allowed us to identify particles with sizes between 1 and greater than 2000 um. The particles that presented greater sphericity, compactness, circularity, solidity, and extension were those present in the size ranges 50–100, 100–200, 200–300, 600–700, 700–800, and 1500–2000, respectively. This shows that the ranges less than 50 um are not spherical. The transparency of the particles across the size range of the study increased with the decreasing size. The DSC technique allowed for the identification of the presence of PET, PP, PE, and PS microplastics after determining their melting points. The confirmation of these families of microplastics was achieved with the use of Py-GC/MS. Pyrolysis allowed the identification of molecules such as benzene, vinyl benzoate, benzoic acid, diphenyl, divinyl terephthalate, ethanediol dibenzoate, and 2-(benzoyloxy) ethylvinyl terephthalate, 2,4,6-trimethyl-1-nonene, 2,4,6,8-tetramethyl-1-undecene, 2,4,6,8,10-pentamethyl-1-tridecene, and 2,4-Dimethyl-1-heptene, styrene, α-methylstyrene 1,3-Diphenylpropane, styrene dimer and styrene trimer, n-pentadec-1-ene which are typical of the pyrolysis of PET, PP, PS, and PE. The wide range of the morphological profile of the microplastics identified in the breathable air in all the study areas of the city of Cartagena shows a high level of contamination for the population and, therefore, this type of research should be an important input to accelerate the development of remedial techniques.

## Figures and Tables

**Figure 1 molecules-28-01042-f001:**
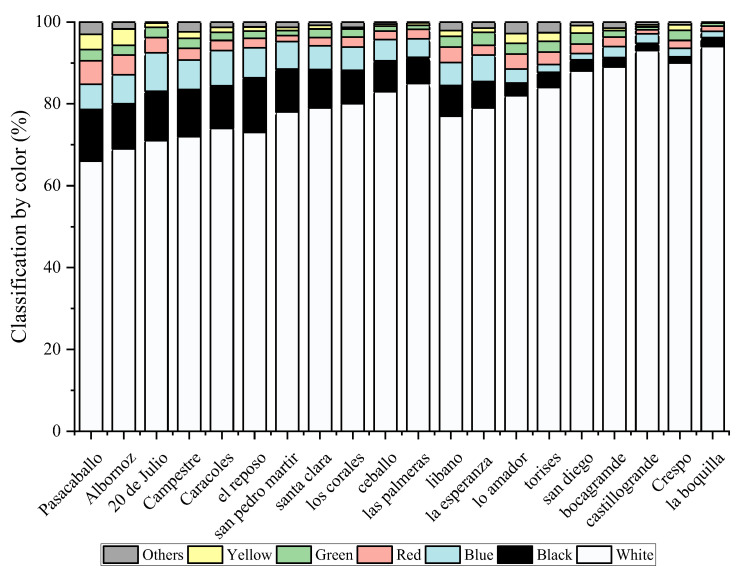
MP composition by color.

**Figure 2 molecules-28-01042-f002:**
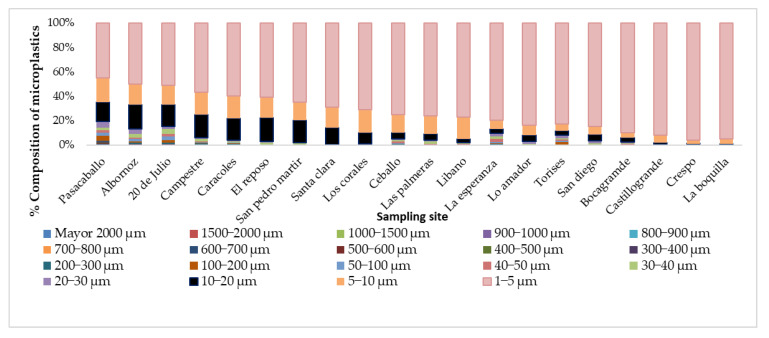
Distribution of the size of the MPs in the different neighborhoods.

**Figure 3 molecules-28-01042-f003:**
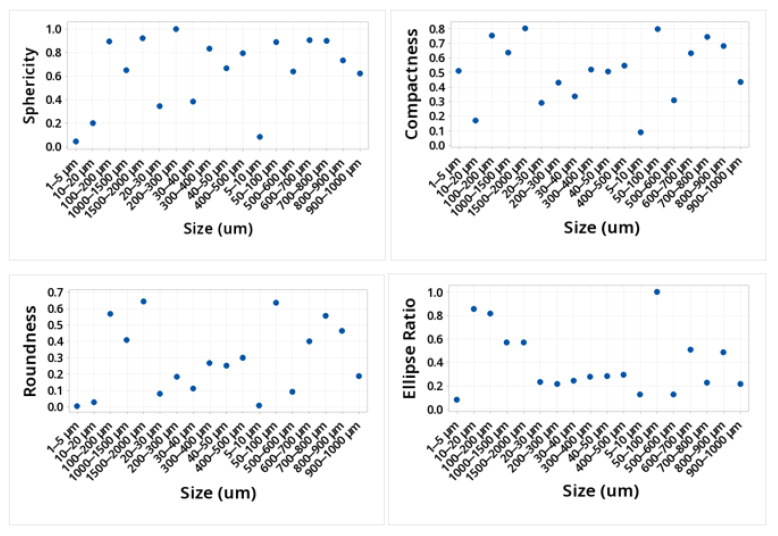
Values of sphericity, compactness, roundness, and ellipse ratio of the MPs identified.

**Figure 4 molecules-28-01042-f004:**
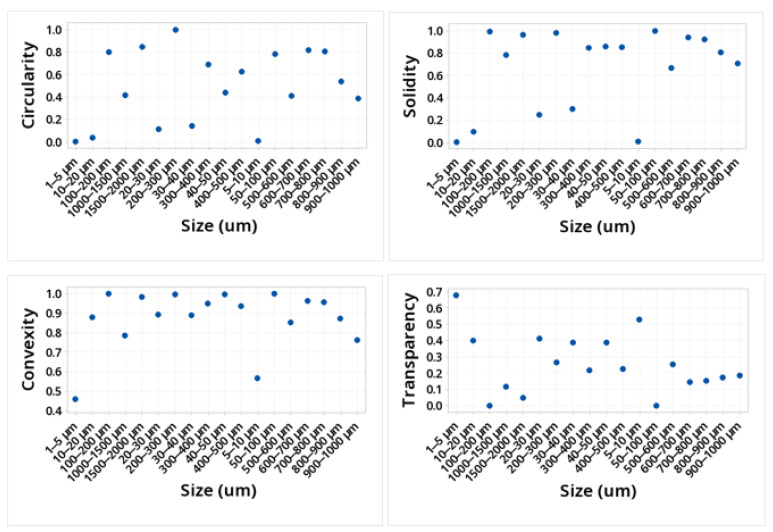
Values of circularity, solidity, convexity, and transparency for the MPs identified.

**Figure 5 molecules-28-01042-f005:**
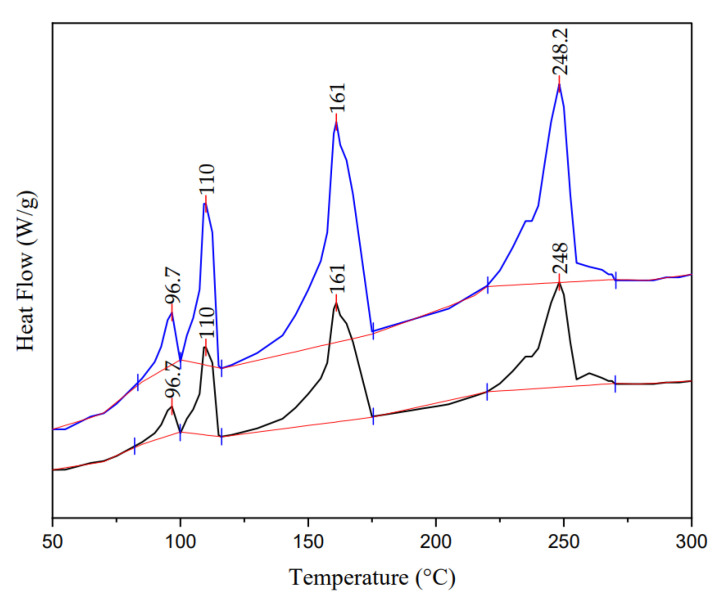
Sample thermogram with MPs mixture.

**Figure 6 molecules-28-01042-f006:**
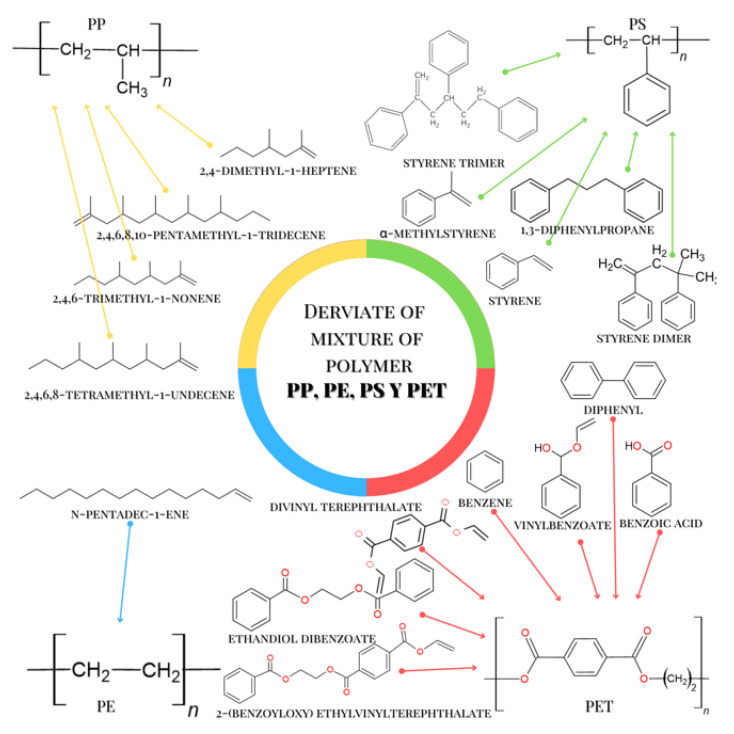
Chemical compounds identified by GC-MS and MPs that originate it.

**Figure 7 molecules-28-01042-f007:**
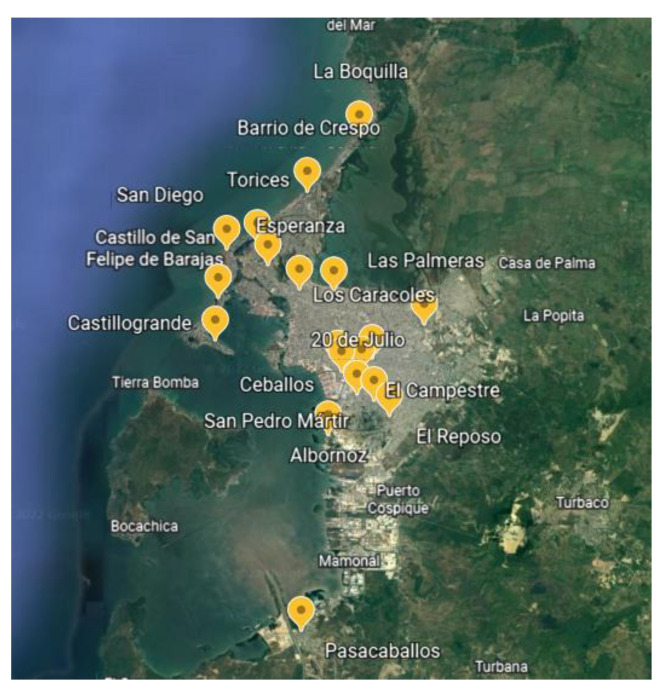
Sampling sites.

**Figure 8 molecules-28-01042-f008:**
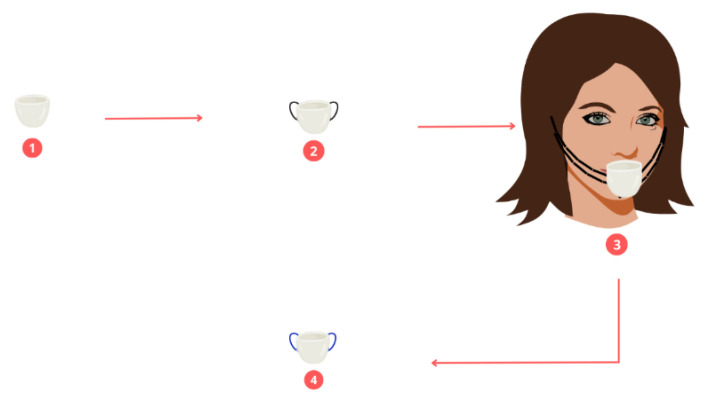
Breathing air sampling.

**Table 1 molecules-28-01042-t001:** Values of morphological parameters for the MPs identified.

Size	Morphological Parameters Identified									
µm	Sphericity	Compactness	Roundness	Ellipse Ratio	Circularity	Solidity	Concavity	Convexity	Extent	Transparency
1500–2000	0.92	0.802	0.642	0.573	0.847	0.964	0.036	0.983	0.912	0.051
1000–1500	0.648	0.638	0.407	0.572	0.42	0.786	0.214	0.785	0.619	0.117
900–1000	0.624	0.436	0.19	0.218	0.39	0.709	0.291	0.763	0.627	0.185
800–900	0.734	0.682	0.465	0.485	0.538	0.807	0.193	0.872	0.657	0.176
700–800	0.898	0.744	0.554	0.231	0.807	0.923	0.077	0.958	0.916	0.153
600–700	0.905	0.632	0.399	0.509	0.819	0.942	0.058	0.965	0.617	0.146
500–600	0.64	0.31	0.096	0.13	0.41	0.67	0.33	0.854	0.511	0.255
400–500	0.792	0.549	0.301	0.299	0.627	0.856	0.144	0.938	0.766	0.229
300–400	0.832	0.519	0.269	0.283	0.692	0.847	0.153	0.949	0.801	0.22
200–300	1	0.432	0.186	0.22	1	0.983	0.017	0.998	0.668	0.267
100–200	0.895	0.754	0.569	0.815	0.801	0.991	0.009	0.999	0.632	0
50–100	0.886	0.796	0.634	1	0.784	1	0	1	0.705	0
40–50	0.664	0.505	0.255	0.286	0.441	0.86	0.14	0.996	0.488	0.389
30–40	0.385	0.337	0.113	0.247	0.148	0.301	0.699	0.891	0.203	0.388
20–30	0.342	0.291	0.085	0.237	0.117	0.248	0.752	0.893	0.168	0.414
10–20	0.199	0.173	0.03	0.854	0.04	0.097	0.903	0.879	0.055	0.4
5–10	0.085	0.089	0.01	0.128	0.01	0.01	0.01	0.57	0.011	0.53
1–5	0.046	0.51	0.008	0.085	0.005	0.005	0.004	0.46	0.004	0.68

**Table 2 molecules-28-01042-t002:** Identification of the MPs in the different neighborhoods.

Neighborhood	Size Range (µm)				
	PP	PE	PET	LDPE	PS
Pasacaballo	1 to >2000				
Albornoz	1 to 2000				
20 de Julio	1 to >2000	1 to 300		1 to >2000	
Campestre	1 to >2000	N/I *		1 to >2000	1 to >2000
Caracoles	1 to >2000	1 to 5 and 500 to >2000		N/I	
Las palmeras	1 to 100				
Líbano	1 to 100	1 to 5		1 to 100	
La esperanza	1 to 900				
Lo amador	1 to 40				
Torises	1 to 400	1 to 40		1 to 400	
El reposo	1 to 100	1 to 5	1 to 100	1 to 5	
San Pedro martir	1 to 500	1 to 5		1 to 500	
Santa clara	1 to 500	1 to 5		1 to 600	
Ceballo	1 to 600	1 to 5		1 to 600	
Los Corales	1 to 600	1 to 5		1 to 600	
San diego	1 to 50	N/I	1 to 50		
Bocagrande	1 to 100	1 to 5	1 to 100		
Castillogrande	1 to 700				
Crespo	1 to 50				
La boquilla	1 to 20				

* Not identified.

**Table 3 molecules-28-01042-t003:** Py-GC-MS calibration information for individual PP, PE, PS, and PET.

Polymer	Marker	m/z	RI	RI (Theoretical)	Equation	R2
PS	2,4-diphenyl-1-butene	208	1729	1749	y = 3.3 × 10^3^x + 1.2 × 10^4^	0.993
PP	2,4-dimethyl-1-heptene	43	837	845	y = 8.5 × 10^3^x + 6.1 × 10^4^	0.993
PE	n-tetradec-1-ene	55	1392	1392	y = 5.4 × 10^2^x + 1.1 × 10^3^	0.996
PET	vinyl benzoate	105	1137	1143	y = 2.7 × 10^3^x − 2.3 × 10^4^	0.878
	divinyl terephthalate	175	1570	1577	y = 3.5 × 10^3^x − 1.0 × 10^4^	0.958

**Table 4 molecules-28-01042-t004:** Chemical compound data obtained by Py-GC-MS when analyzing polymer blends.

Groups	Identified Compounds	RI	Qualification Ions (m/z)	Polymers
1	benzene	–	51, 52, 78	PET
	vinylbenzoate	1137	77, 105, 148	PET
	benzoic acid	1214	77, 105, 122	PET
	diphenyl	1379	76, 153, 154	PET
	divinyl terephthalate	1567	76, 104, 175	PET
	ethandiol dibenzoate	2190	77, 105, 227	PET
	2-(benzoyloxy) ethylvinylterephthalate	2662	105, 149, 297	PET
2	2,4,6-trimethyl-1-nonene	1079	41, 69, 168	PP
	2,4,6,8-tetramethyl-1-undecene	1307	43, 69, 154	PP
	2,4,6,8,10-pentamethyl-1-tridecene	1527	43, 69, 196	PP
	2,4-dimethyl-1-heptene	837	43, 70, 126	PP
3	styrene	898	51, 78, 104	PS
	α-methylstyrene	982	103, 117, 118	PS
	1,3-diphenylpropane	1659	92, 105, 196	PS
	styrene dimer	1726	91, 104, 208	PS
	styrene trimer	2510	91, 117, 312	PS
4	n-pentadec-1-ene	1497	43, 55, 210	PE

## Data Availability

Not applicable.

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
