# Peer review of "Characterization of the Morphological and Chemical Profile of Different Families of Microplastics in Samples of Breathable Air"

_molecules, 2023, doi:10.3390/molecules28031042_

Round 1

Reviewer 1 Report

Line 77, “breathable air samples are taken in 19 neighborhoods of the city of Cartagena, Colombia.” Here is “19 neighborhoods”, while others are all “20 neighborhoods” in the manuscript.

Since temperature and humidity could affect the movement and transport of MPs in the environment, the authors should provide the data of the temperature and humidity in these sampling sites.

Line 99-103,I still do not understand how breathing air was sampled. Can the authors provide the schematic picture to present how to sample the breathing air?

Line 137, what does “He was used as carrier gas at 1.0 mL min-1” mean?

Line 165-166, “Those that were identified had a particle size greater than 800 m, so they could be seen.” Greater than 800 m or 800 µm?

Figure 4. Values of morphological parameters of the MPs identified. All the x-coordinate are “Zise” or “Size”?

Line 312-322, this paragraph is not in English!

Where is the Table 4 cited in the main text?

Where is Figure 6 cited in the main text? What does these structure in Figure 6 mean for your results?

Line 328-337, these sentences are not your Conclusions.

There are some recent literatures about airborne microplastics should be referenced. E.g., Indoor microplastics and bacteria in the atmospheric fallout in urban homes. Sci. Total Environ. 2022, 852, 158233.

Author Response

1-Line 77, “breathable air samples are taken in 19 neighborhoods of the city of Cartagena, Colombia.” Here is “19 neighborhoods”, while others are all “20 neighborhoods” in the manuscript.

R/Very kind for your review. We make the correction.

2- Since temperature and humidity could affect the movement and transport of MPs in the environment, the authors should provide the data of the temperature and humidity in these sampling sites.

R/Very kind for your review. The temperature during the sampling periods oscillates between 27 °C and 30 °C. The reactive humidity varied between 80 and 83%. now we have placed the values of temperature and humidity in numeral 2.2.

3- Line 99-103,I still do not understand how breathing air was sampled. Can the authors provide the schematic picture to present how to sample the breathing air?

R/Very kind for your review. We have posted an image that will help you understand how the sample of microplastic in the breathable air was taken.

4- Line 137, what does “He was used as carrier gas at 1.0 mL min-1” mean?

R/Very kind for your review. This refers to Helium. Helium is the gas in the chromatograph. This gas is responsible for transporting the sample into the chromatographic column. Now we have better written this paragraph in the document.

5- Line 165-166, “Those that were identified had a particle size greater than 800 m, so they could be seen.” Greater than 800 m or 800 µm?

R/Very kind for your review. We make the correction.

6- Figure 4. Values of morphological parameters of the MPs identified. All the x-coordinate are “Zise” or “Size”?

R/Very kind for your review. We make the correction.

7- Line 312-322, this paragraph is not in English!

R/Very kind for your review. We make the correction.

8- Where is the Table 4 cited in the main text?

R/Very kind for your review. From line 313 to 346 we quote Table 4.

9- Where is Figure 6 cited in the main text? What does these structure in Figure 6 mean for your results?

R/Very kind for your review. In lines 344 to 350 we refer to Figure 6. This Figure 6 shows the condensed structure of the PET, PP, PE, PS polymers and also of the most representative molecules that were identified after the pyrolysis of microplastics. With the information in this Figure 6, we can see that the molecular structure of the chemical species have common functional groups with each microplastic and in this way we can clearly see that the origin of each molecule identified by mass spectrometry originates from pyrolysis. of each specific polymer.

10- Line 328-337, these sentences are not your Conclusions.

R/Very kind for your review. We make the correction.

11- There are some recent literatures about airborne microplastics should be referenced. E.g., Indoor microplastics and bacteria in the atmospheric fallout in urban homes. Sci. Total Environ. 2022, 852, 158233.

R/Very kind for your review. This scientific publication is very good. We have read it and now quoted in this investigation.

Reviewer 2 Report

Dear authors,

I appreciate the significance of content of your work, even if the quality of presentation is in my opinion low.

There are many errors:

-Introduction line 77 you wrote 19 neighborhood but than you wrote 20;

- Results and discussion line 166 you wrote 800 m, m stands for?

- figure 3 refers to Maps dimensions but you didn't put the unit measurement;

- figure 4 has a very low quality, is it difficult to read;

- line 240: missing a dot after "them";

- line 293: missing bracket in front of "between";

- line 306: missing capital letter to "this" an to "these" in the next sentence;

- figure 4 caption is in spanish

Some technical concern:

In paragraph 2.2.1 you wrote 50 ml solution of Nile Red was added, but at which concentration? And how was prepared Nile red solution? In which solvent was dissolved? 

For what investigation did you use Nile red? Did you study fluorescence?

Author Response

1- Introduction line 77 you wrote 19 neighborhood but than you wrote 20;

R/Very kind for your review. We make the correction.

2-  Results and discussion line 166 you wrote 800 m, m stands for?

R/Very kind for your review. We make the correction.

3- figure 3 refers to Maps dimensions but you didn't put the unit measurement;

R/Very kind for your review. We make the correction.

4- figure 4 has a very low quality, is it difficult to read;

R/Very kind for your review. We make the correction.

5- line 240: missing a dot after "them";

R/Very kind for your review. We make the correction.

6- line 293: missing bracket in front of "between";

R/Very kind for your review. We make the correction.

7- line 306: missing capital letter to "this" an to "these" in the next sentence;

R/Very kind for your review. We make the correction.

8- figure 4 caption is in Spanish

R/Very kind for your review. We make the correction.

Some technical concern:

9-In paragraph 2.2.1 you wrote 50 ml solution of Nile Red was added, but at which concentration? And how was prepared Nile red solution? In which solvent was dissolved? 

R/Very kind for your review. The concentration of the red null prepared was 100 ppm and acetone was used as the diluting solvent. We have now placed this information in the text.

10-For what investigation did you use Nile red? Did you study fluorescence?

R/Very kind for your review. We use Nile Red (also known as Nile Blue Oxazone) as a color indicator for microplastics. given its nonpolar nature. This indicator made it possible to indicate in the first instance if there were microplastics present and to be able to easily see them on the equipment screen. After knowing that the sampling technique could work for us, we then stopped using the indicator in the following samples, and thus differentiate microplastics of different colors, such as those that were identified in this investigation.

Round 2

Reviewer 1 Report

All my questions and concerns have been addressed.

Reviewer 2 Report

Dear authors,

thank you for improving your work.

Line 111: you wrote "red null", is it a typo error for Nile red?